# Qualitative Field Observation of Pedestrian Injury Hotspots: A Mixed-Methods Approach for Developing Built- and Socioeconomic-Environmental Risk Signatures

**DOI:** 10.3390/ijerph17062066

**Published:** 2020-03-20

**Authors:** Nadine Schuurman, Blake Byron Walker, David Swanlund, Ofer Amram, Natalie L. Yanchar

**Affiliations:** 1Department of Geography, Simon Fraser University, 8888 University Drive, Burnaby, BC V5A 1S6, Canada; dswanlun@sfu.ca; 2Institut für Geographie, Friedrich-Alexander-Universität Erlangen-Nürnberg, Wetterkreuz 15, 91058 Erlangen, Germany; blake.walker@fau.de; 3Spokane Health Education and Research Building Third Floor, P O Box 1495, Spokane, WA 99210-1495, Canada; ofer.amram@wsu.edu; 4Pediatric General Surgery, Alberta Children’s Hospital, 28 Oki Drive NW, Calgary, AB T3B 6A8, Canada; natalie.yanchar@ahs.ca

**Keywords:** pedestrian injury, geographic information systems, mapping, hotspots, socio-economic status, mixed methods

## Abstract

Road traffic injuries constitute a significant global health burden; the World Health Organization estimates that they result in 1.35 million deaths annually. While most pedestrian injury studies rely predominantly on statistical modelling, this paper argues for a mixed-methods approach combining spatial analysis, environmental scans, and local knowledge for assessing environmental risk factors. Using data from the Nova Scotia Trauma Registry, severe pedestrian injury cases and ten corresponding hotspots were mapped across the Halifax Regional Municipality. Using qualitative observation, quantitative environmental scans, and a socioeconomic deprivation index, we assessed hotspots over three years to identify key social- and built-environmental correlates. Injuries occurred in a range of settings; however, clear patterns were not observed based on land use, age, or socio-economic status (SES) alone. Three hotspots revealed an association between elevated pedestrian injury and a pattern of geographic, environmental, and socio-economic factors: low- to middle-SES housing separated from a roadside attraction by several lanes of traffic, and blind hills/bends. An additional generalized scenario was constructed representing common risk factors across all hotspots. This study is unique in that it moves beyond individual measures (e.g., statistical, environmental scans, or geographic information systems (GIS) mapping) to combine all three methods toward identifying environmental features associated with pedestrian motor vehicle crashes (PMVC).

## 1. Introduction

### 1.1. Pedestrian Injury

Injury represents a fundamentally global public health burden, although it is also highly preventable. Estimates suggest that in 2013, between 942 and 993 million people required health care due to a sustained injury [1]. That same year, 4.8 million individuals had their lives taken by injury [1]. In terms of years of life lost, injury resulted in nearly 260 million years of life lost globally in 2012 alone [2]. More specifically, road traffic injuries constitute a significant component of the global burden of injury, as they end approximately 1.35 million lives every year [3]. Such injuries are not without a significant economic impact: according to the World Health Organization (WHO), road traffic injuries reduce national gross domestic product by 3% on average [3]. Clearly, injuries on the roadway present a significant problem for public health. 

### 1.2. Objectives

Building on our previous studies of injury hotspots [4,5], this study sought to develop a mixed methods approach to the assessment of risk factors associated with pedestrian injury hotspots. The approach integrates GIS mapping, detailed environmental scans, and statistical analysis of socio-economic status indicators. The goal was to combine qualitative and quantitative assessments to develop a greater understanding of high-risk settings that can ultimately be used for an injury prevention policy in Halifax, Nova Scotia. Risk has a complex signature and we argue that it is valuable to combine qualitative observations with the assembled quantitative data in order to characterize risk in specific settings. The calculation of socio-economic status (SES) and the geographical and environmental correlates identified in the study are generalizable to other urban areas, but each setting will have unique combinations of the correlates that must be assessed in situ.

### 1.3. Mixed Methods

The exclusive use of quantitative methods (specifically statistical modelling) to analyze risk factors suffers from several weaknesses when applied to injury studies. A reliance on large sample sizes/populations to achieve adequate statistical power biases studies of smaller locales towards the null hypothesis, almost certainly leading to important patterns and risk factors being rejected due to inadequate statistical significance [6,7,8,9]. A traditional hypothesis testing (or otherwise *p*-value or confidence-interval-focused) approach additionally relies on assumptions of underlying distributions to assume multiple samples from a consistent population (a particularly perilous problem when using geospatial models), which, well suited to inferential modelling of highly-controlled experimental conditions [10], falls short of accounting for the non-parametric nature of models. However, the use of statistical tools to evaluate injury data does present important inroads to disentangling patterns of injury and identifying risk factors. It may therefore be argued that drawing upon both quantitative and qualitative methodologies better equips researchers and analysts to detect and characterize risk environments for complex socioeconomic and built-environmental phenomena.

Accordingly, there has been a burgeoning of mixed methods research in the health and medical domain, as the value of using two epistemic angles to triangulate possible reasons or rationales for convergence of health incidence has been realized [11,12,13,14]. While a quantitative approach affords the ability to identify clusters of disease incidence, the unique local conditions that lead to their formation are harder to uncover. Qualitative research in an ideal situation does not bring a priori assumptions about the conditions that lead to such clusters. Rather, qualitative research examines through an idiographic approach the conditions that exist in situ that facilitate clusters of disease incidence, in this case pedestrian injury. By utilizing the strengths of both methodologies through a mixed method approach we are able to craft unique risk signatures for pedestrian injury hot spots that would otherwise be missed if either qualitative or quantitative methods were used on their own. Were this research to focus on quantitative measures alone, we could identify the hotspots and conduct the environmental scans, but it would be difficult to explain why the hotspots were in that location and how they were related to the environmental correlates. Likewise, qualitative research alone might be able to use inductive reasoning and local knowledge to predict possible hotspots, but researchers would have no way to corroborate their suspicions regarding possible causality. The combination of the two approaches permits the unique methodology presented here. 

### 1.4. Risk Factors

Previous studies have identified risk factors for pedestrian injury related to motor vehicles (PMVC) including those related to vehicle traffic, infrastructure, and socio-economic contexts. These are discussed below and are the basis for the geographical framework used in this study.

**Traffic Volume:** Traffic volume, whether represented directly by measures such as vehicle miles travelled or indirectly by arterial road presence, is one of the strongest risk factors associated with PMVCs. Research has consistently shown that areas with higher traffic volume result in more PMVCs, as well as higher injury severities [15,16,17,18,19,20]. This effect is not necessarily distributed uniformly in space. For example, traffic volume is a significant risk factor at intersections but not mid-block [21]. This may be because high traffic volumes introduce more complexity at intersections, rather than between them. 

**Road Width:** Closely related to traffic volume is road width, which can be examined as the number of lanes as well as the width of those lanes. With regard to the former, numerous studies have found that the number of lanes on a road is significantly associated with increased risk of PMVC [15,18,22,23]. This is due to the added distance that pedestrians are required to cross, thereby increasing their exposure to traffic. Wide lanes also increase PMVC risk [24]. Godley et al. [24] suggest that this is because narrow lanes demand an increased steering workload that causes drivers to travel at slower speeds. This correlates with motor vehicle collision (MVC) research that has demonstrated increased MVC rates with wide lanes [25,26,27]. Finally, the addition of a divider in the center of the road and the addition of a painted center-hatch both decrease PMVC risk, likely by attenuating drivers’ speed estimates, resulting in slower vehicle speeds and decreased crash risk [22,24].

**Intersections:** The design of intersections also affects PMVC risk. It is widely agreed that four-legged intersections are more dangerous for pedestrians than their three-legged counterparts [15,18,28,29], and that adding more approaches only exacerbates PMVC rates [18,30]. While in some cases this may be a reflection of traffic volume, adding more approaches to an intersection may increase its complexity, making it more dangerous for pedestrians.

**Crossing Infrastructure:** The installation of crossing infrastructure, such as signals or crosswalks, can promote pedestrian safety. Pedestrians crossing without a signal or against a signal have been clearly shown to be at higher risk of PMVC, and over half of pedestrians killed are not in a crosswalk [17,31]. The absence of signals in rural areas is also strongly associated with more crashes; pedestrians crossing non-signalized intersections on major routes are four times more likely to be severely injured or killed than elsewhere on the route [17,22,32].

**Street Parking:** Street parking has a demonstrated effect on pedestrian injury as it modifies pedestrian behavior. Morency et al. [28] found that pedestrian injury rates are higher at intersections with cars parked within five meters of their perimeter. Dommes et al. [33] found that pedestrians were more likely to violate the signal at intersections if cars were parked legally around them, yet if cars were parked *illegally* surrounding the intersection, pedestrians would instead exercise more caution when deciding to cross [34]. These findings may suggest that legal parking lures pedestrians into a false sense of security or trust in the infrastructure to protect them.

**Alcohol Outlets:** Pedestrian injury has been associated with proximity to alcohol-serving establishments, possibly due to increased risk of intoxication by either drivers or their victims [7,35,36]. Many studies have demonstrated that pedestrians who are intoxicated sustain injuries at a greater rate and severity than those who are not [23,31,37,38]. Driver intoxication also increases the risk of deadly injury for pedestrians [17,32].

**Land Use Mix:** Land use has a well-documented effect on PMVCs. For instance, residential areas have lower rates of PMVCs [15,30,39]. Conversely, commercial, industrial, and mixed land-use areas sustain higher rates of PMVCs [15,20,21], including a greater risk of severe injuries or fatalities [17,39,40]. This is likely related to increased traffic volume, speed, and overall traffic pattern complexity of these areas. In addition, land use mix can influence walking behaviors, thus increasing—or conversely, decreasing—exposure to risk. 

**Public Transportation:** The presence of public transportation infrastructure appears to have a mixed effect on pedestrian injury. Some have found the presence of transit stops to be associated with an increase in the number of crashes in a given area [15,28,30,41], while others have found that transit access is associated with significantly reduced injury severity [31]. It is possible that areas with transit attract more pedestrian traffic, thereby resulting in more PMVCs, but that collisions in these areas occur at a lower speed due to drivers being alert to the greater pedestrian presence and thereby reducing the severity of injuries. Ultimately, the spread in results associated with studying the links between transit and risk of pedestrian injuries may be explained by other unobserved factors in the built urban environment.

**Socioeconomic Status:** Unsurprisingly, socioeconomic status is strongly associated with PMVC rates. Studies have linked low area-level educational attainment to higher PMVC rates [16,18,20,29,42,43] as well as increased severity of subsequent injuries [32]. 

**Age:** Age has a well-established association with the risk of severe injury in pedestrian-vehicle collisions. Numerous studies have found that older pedestrians are at higher risk for sustaining more severe or deadly injuries when struck by vehicles [15,31,32]. However, the rate at which older pedestrians are injured is less clear, with some studies finding rates similar to or lower than the general population [31,42], and others finding far higher rates [44]. Similarly, there are conflicting findings with regard to the behavior of older pedestrians. Some studies report that older pedestrians comply more often with crossing signals and are generally more cautious [33,45]. Others suggest that older pedestrians may have poorer judgment related to slower walking speeds [46] while relying more heavily upon crossing infrastructure [33]. Dommes et al. [47] found that functional age-related declines affected a variety of factors in older pedestrians, suggesting that they “may overestimate their speed of walking, misperceive the time-to-arrival of approaching cars, and have difficulty in efficiently compensating for their perceptual and attentional declines when crossing a two-way street, especially when traffic is approaching at a high speed.” Finally, there may be differences in walking ability levels among the cohorts in the different studies. 

Younger pedestrians are also at an increased risk. Moudon et al. [32] found that children under the age of four have almost three times the general population’s risk of severe injury or death within cities due to PMVCs. Dai’s [44] analysis of risk factors in pedestrian injury returned the highest injury rates for those under the age of 14, and Chakravarthy et al. [42] found that areas with a greater proportion of residents under the age of 14 suffered more crashes. On the other hand, LaScala et al. [16] found that areas with higher proportions of children had lower injury rates. Ultimately, despite conflicting findings with regard to the specific type and degree of vulnerability, it is clear that pedestrians at either end of the age spectrum are more vulnerable to severe injury when involved in a crash compared to those in the middle. As with the variables discussed above, exposure is a key component of risk but often not accounted for. 

## 2. Materials and Methods 

There were two components to this mixed methods study. The first component consisted of a methodical approach to data collection by mapping hotspots using known risk factors (from the literature) to compile an environmental scan survey instrument, and finally carrying out the environmental scans with a team of four. The second component of the study emphasized field work to understand primary patterns that emerged from the data. We combined socio-economic status data (SES) with field observations to understand why particular patterns emerged. The protocol for the methods is illustrated in Figure 1.

### 2.1. Data

Data were acquired from the Nova Scotia Trauma Registry which maintains a comprehensive population-based database of all cases of serious injuries taken to hospital in the Canadian province of Nova Scotia (Injury Severity Score >11, requiring trauma team activation and/or resulting in in-hospital death). All individuals in the database sustaining a severe pedestrian injury within the Halifax Regional Municipality (HRM) of Nova Scotia from 1 June 2002 to 31 March 2015 were included. Variables included in this analysis were incident date/time, location recorded by ambulance GPS, patient age and sex, and the injury severity score (ISS) recorded at hospital. 

Research ethics approvals were obtained from IWK Health Centre Research Ethic Board (Project #1012595) and from Simon Fraser University Office of Research Services (Study #2013s0873).

### 2.2. Geographical Analysis

All pedestrian injuries were geocoded based on the GPS location data and were mapped using ESRI ArcGIS® software. Kernel density estimation (KDE) was used to identify the chief hotspots using a bandwidth of 500 m (which approximates the average city block). KDE was selected based on its use in previous injury studies [7,48,49] and because it indicates location hotspots (based on hue) without revealing individual patient locations. 

### 2.3. Environmental Scans

We used a detailed literature review to identify chief correlates of PMVC to include in the survey instrument. The instrument was first used in a study conducted by the authors in Vancouver, BC [7]. Attributes included in the walking in-person environmental scans were: long block (Yes or No); bus stop (Y or N); curb parking (Y or N); cross-walk (Y or N); obstruction (Y or N); signage (high, medium, low, subjectively measured); lanes of traffic; L/R turn bans (Y or N); bars (Y or N); retail (Y or N); school (Y or N); traffic calming measures (Y or N); median (Y or N); exclusive turn lane (Y or N); and speed limit. In addition, we took detailed notes, photos, and videos at each hotspot. The environmental scan framework is illustrated in Figure 2 below.

Over the course of three years, we returned on three occasions to each of the hotspots in an attempt to understand inductively why certain combinations of correlates led to higher injury counts. It is possible that the built urban environment could have changed significantly between injuries at some locations (e.g., new signage or streetlights). However, this risk was mitigated by our return to each hotspot to conduct new scans over the course of three years.

Hotspots were identified from the initial analysis and mapping. For the purposes of this study, we focused on the top ten hotspots, for which three separate environmental scans were conducted, once in winter and again in summer. The results were enumerated based on the scores of four reviewers. In addition, land use at each hotspot (retail, recreational, or residential) was recorded. Notes were also taken on the traffic environment (volume, speed, pedestrian protection, presence of blind corners, and confusing/uncommon roadway configuration). Interestingly, there was strong agreement among the environmental scans team when counting the features and in recording notes.

In addition, socio-economic status (SES) was calculated at the Dissemination Area (DA) level. The DA is the smallest unit for which Statistics Canada compiles and publicly publishes detailed census data. Typically, a DA contains 400 to 700 people. DAs were designed to maximize the following attributes in order better characterize neighborhoods: temporal stability (borders not subject to frequent change); minimum population (i.e., greater resolution); uniformity of attributes; compact shape; and intuitive boundaries [50]. As such, they are well suited for characterizing SES at the population level. The Vancouver Area Neighbourhood Deprivation Index (VANDIX) metric was used to calculate SES (and, by extension, deprivation) for each DA in the HRM [51,52]. The VANDIX was developed explicitly for calculating deprivation across Canada. The measure includes variables associated with social capital (e.g., home ownership; living alone) as well as standard measures of income and education. The SES value was separately calculated for the DA containing each unique hotspot. These results were then compiled with environmental scan results to characterize contributing factors and contexts related to each PMVC hotspot.

Several in-person and remote meetings were held with all authors/researchers to discuss the range of results from the various analysis tools. These meetings were used to critically examine the results from each method in order to synthesize the findings. 

## 3. Results

The results are reported in a range of formats that reflect the mix of GIS (geographical), environmental scan, and SES methods used to assess risk factors at PMVC hotspots. Figure 3 presents a broad overview of the results. The figure illustrates that, at first glance, it is difficult to discern consistent patterns across the ten hotspots. However, injuries were found to have occurred in a range of land use and SES settings. Further quantitative results combined with a qualitative assessment are required to understand the risk factors associated with individual hotspots and intersections. 

Table 1 (below) provides greater clarification of the map view in Figure 3. Here we see that age-related risk is consequential; in our sample, persons aged 11 to 30 years bore the greatest risk but with significant risk also amongst those 51 to 60 years. There is a slightly elevated risk for males over females. In terms of SES, those at both ends of the deprivation spectrum bore the most injuries, while those in the mid-SES range suffered fewer injuries. Table 2 lists frequency counts for each risk factor.

Despite the utility of these quantitative summaries, ultimately we were best able to characterize the commonalities among the most risky intersections through qualitative discussion and examination of field notes, photos (such as Figure 4), and video. We were aided by the local knowledge held by one of the authors. We observed that three hotspots shared common risk factors: low- to middle-SES housing separated by at least four lanes of traffic from an attractive feature such as fast food outlets, retail establishments, a casino, or a sport complex. In each instance, these busy roads featured long blocks, a blind hill and/or bend, and no median or traffic calming measures.

Figure 5 illustrates a more generalized scenario that was commonly encountered across all ten hotspot locations. The net effect was that pedestrians were forced to cross four or more lanes of heavy traffic with regularity, often to reach either a bus stop or roadside attraction. There were typically no crosswalks, medians, or traffic calming measures to improve pedestrian safety. Long blocks, exclusive turns, and high signage were common. Low- to middle-SES housing as well as blind hills and/or bends were less frequent but were agreed to be important factors when present.

## 4. Discussion

This study was unique in that it sought to move beyond individual measures (e.g., statistical analysis, environmental scans, or GIS mapping) to characterize environmental features associated with higher risk of PMVC. Rather, it incorporated all of these measures with the net effect that patterns were revealed only by integration of all modalities of reporting. As described in the results section above, a specific pattern emerged in three of the ten PMVC hotspots. In each of these, low- to middle-SES housing was separated from a major roadside attraction by four or more lanes of traffic and blind bends and/or hills. In addition, no median, crosswalk, or traffic calming measures were present, and blocks were long. This unique combination of geographical, environmental, and SES factors was associated with higher rates of pedestrian injury. A more generalized scenario that is representative of common risk factors across all ten hotspots was also described and is again represented in Figure 5.

What sets this study apart was its use of multiple analytical methods to assess risk factors, as well as ultimately utilizing an informal qualitative assessment to identify these patterns. 

Of course, each urban area will feature unique constellations of risk factors. For this reason, the integration of multiple analytic methods, united by a final qualitative assessment based on local knowledge combined with quantitative results is of particular utility and is strongly recommended for future research [53].

This research is decidedly qualitative despite being cloaked in quantitative clothing, which begs the question of asking what makes research qualitative. Certainly there is discussion amongst scholars as to what constitutes *qualitative* [54,55]. For the purposes of this study, we have used our inclusion of observational study and inductive pattern recognition as a measure of qualitativeness [56]. This was augmented by returning to the study site on three occasions as we sought to understand why the patterns of injury differed so much from a previous study we conducted in Vancouver using the same methods [4]. Returning to the same site over a period of time to try and unravel patterns is a form of ethnography that is well suited to understanding why some intersections are manifestly high risk while others are not [57]. Health research, especially spatial epidemiology, has not traditionally embraced qualitative approaches [58] and we see an opportunity here to both expand the purview of qualitative health research as well as the methods. Traditional epidemiology relies upon large numbers and statistical measures of certainty to establish rigour, however, there are methods to corroborate qualitative methods based on researcher reflexivity and justification of methods [59]. If the final goals of using a qualitative approach is to extend understanding of a situation and find policy solutions to a problem [60], then we believe that this paper achieves this based on its qualitative and inductive approach.

Moreover, this study also stresses the importance of environmental scans and field work—including the use of field notes, photography, and video—when conducting this type of research, as our socio-economic results indicated that high SES DAs included almost as many injuries as low SES DAs. While counterintuitive, our environmental scans provided some explanation for these results, such as the presence of shelters that house populations often not captured by the census. In addition, we relied on part of our team having local knowledge that allowed us to more easily identify key artifacts such as the men’s shelter. Therefore, we encourage others to critically situate their findings using qualitative-interpretive environmental scans [61]. In other words, we suggest that researchers not only identify environmental correlates but seek to explain their unique configurations—especially when those environmental configurations repeat, suggesting greater combined risk (see, for example, Figure 5 above).

Of special interest is that the results of this study contrast starkly with previous studies conducted with a similar dataset in Vancouver, Canada [4,5]. In the 2009 study [4], proximity to alcohol-serving establishments was the greatest risk factor for increased PMVCs. The 2011 study [5] demonstrated that pedestrian violations of crossing laws were high in areas with a higher density of alcohol-serving establishments and lower in areas without bars. Conversely, in the HRM, only two of the top ten hotspots had alcohol-serving establishments nearby. In Halifax, the attractants (and therefore risk factors) were related to sports, recreation, and retail. 

Ultimately, every hotspot is unique. There are, however, shared characteristics and combinations of features between them. We are able to derive both specific and generalized profiles of high-risk settings, comprising combinations of built-environment features and socio-economic characteristics. Assessing these composites is, perhaps, the key to understanding patterns of risk in urban areas. It may be that the combination of unique settlement patterns, geography, SES profile, transportation infrastructure, and by-laws lead to unique risk signatures for different areas. Finally, the risk signature must be influenced by volume of pedestrian traffic. Cinnamon et al. [5] used counters to measure pedestrian traffic at hotspots at various times of day. Certainly, this framework could be expanded to include traffic counts, although counts are capricious as they are influenced by extraneous factors such as weather. 

If indeed each city constitutes a unique risk signature, then this is the key to assessing and remediating risk. Recognition of the existence of unique urban risk signatures requires individual assessment based on known risk factors [4]. In other words, the literature and previous environmental scans can provide a qualitative parameterization for environmental risk assessment for any given urban area. However, the unique risk architecture for a place will emerge only as a result of environmental scans combined with registry data for pedestrian injury. In this study, we used an environmental scan matrix that had been compiled for our 2009 Vancouver study [4]; yet, when combined with SES data, completely different risk profiles emerged.

Within the risk composites that emerge for PMVC, there will always be conflicting data. For instance, we found that exclusive turn lanes were associated with a higher number of pedestrian/motor vehicle injuries. While the data indicate that exclusive turns limit injuries in the context of MVCs, this may not be the case with respect to PMVC. Alternatively, exclusive turn lanes may indicate higher traffic volumes and thus increased complexity and risk for pedestrians. Likewise, injury hotspots in the HRM were associated with long blocks but not bars, contradicting the 2009 study in Vancouver [4]. Additionally, the VANDIX scores calculated in this study were relative to Halifax (i.e., standard scores against the HRM mean VANDIX score), resulting in higher injury frequencies in neighborhoods at either end of the SES spectrum compared to those in the middle. This unexpected result prompted us to conduct a brief sensitivity analysis using standard scores against the Nova Scotia mean and the Canada mean, each of which showed injuries being far more frequent in the highest SES neighborhoods. As such, we reported our results relative to the HRM, but this underscores the importance of spatial scale—as well as the importance of assessing unique risk signatures associated with urban areas.

This study was limited in its ability to statistically infer risk factors due to limited sample size. The use of census data to quantify local socio-economic deprivation at injury hotspots requires an acknowledgement that the injured persons may not be residents of the area in which they were injured. Theoretically, higher density neighborhoods are more likely to host clusters simply as a result of more people walking. We did not find hotspots were in high density residential neighborhoods, though two were in high traffic commercial zones.

The results of this study do, however, suggest that the built environment and its effect on pedestrian injury risk varies geographically, such that risk factors identified in a given neighborhood, city, or region may not necessarily hold true in other settings. Accordingly, local knowledge and expertise is vital to developing a suitably nuanced understanding of patterns of injury and potential responses towards reducing its burden.

Moreover, given that we were working with a 14-year span of data, it is possible that some intersections underwent substantial or partial physical change over the course of the study period, especially those in commercial zones. We expect that the level of commercial infrastructure may have intensified in those areas, however, we are confident that the residential areas included in the scans were relatively static.

Unique urban risk signatures are significant due to their ability to inform local policy. Since each city has its own structure and local conditions that affect PMVC risk, any attempts to mitigate PMVCs should take into account this structure and local conditions. This study, therefore, utilized a variety of methods that included local knowledge and environmental scans. In this case, we can strongly recommend the installation of crosswalks or pedestrian overpasses in several of the hotspots that were studied. Additionally, given that nearly half of the PMVCs we studied occurred in areas of low SES or were proximal to low- to middle-SES housing, our research once again illustrates that injury is related to social determinants of health. Therefore, in addition to the numerous other benefits of increased welfare, policies that attempt to reduce deprivation and inequality will also likely help to reduce the burden of injury. 

## 5. Conclusions

This study used a mixed-methods approach to identify the built-environment factors that contribute to PMVCs in the HRM. More specifically, ten hotspots were identified based on data from the Nova Scotia Trauma Registry, each of which were subject to environmental scans in both summer and winter. Additionally, socio-economic deprivation using the VANDIX was calculated for the areas surrounding the hotspots. Together, the environmental scans and SES data were used to create a unique local risk signature for PMVCs in the HRM. Contributing factors to risk included roads with four wide lanes of traffic across hills and blind bends, the presence of low- to middle-SES housing on one side of the road and a roadside attraction on the other, and the absence of crosswalks, medians, or traffic calming measures. As well, nearly half of PMVCs involved victims aged 11–30 years old. This local risk signature may be significant for assessing and mitigating risk in the future. 

Local knowledge and expertise were consulted throughout this process, and on-the-ground environmental scans provided a more nuanced qualitative understanding of what contributed to these hotspots. Accordingly, we emphasize the importance of going beyond strictly data-based understandings of injury risk and towards more mixed-methods approaches that recognize the uniqueness of cities and the value of on-the-ground research. 

## Figures and Tables

**Figure 1 ijerph-17-02066-f001:**
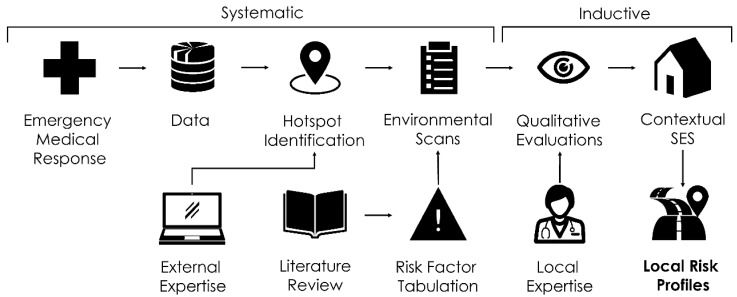
Methods for the study including framework for assessing risk profiles for pedestrian motor vehicle crashes (PMVC).

**Figure 2 ijerph-17-02066-f002:**
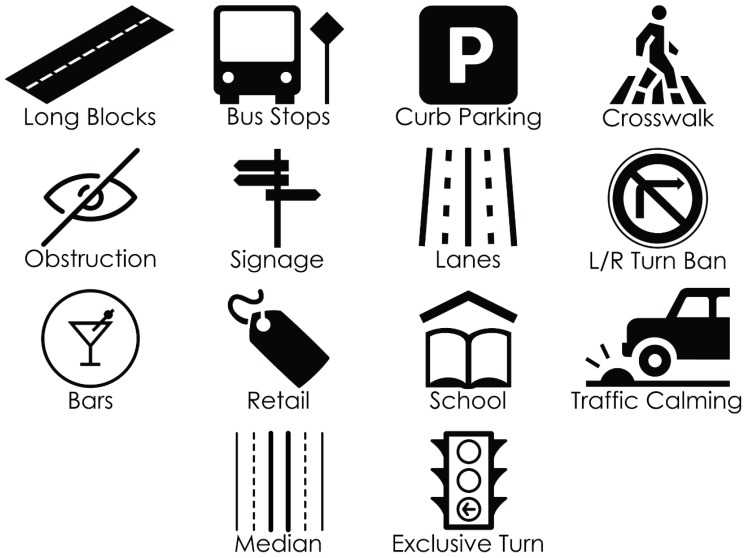
A visual representation of the environmental artifacts included in the survey instrument used during our environmental scans. Each of these environmental elements was selected based on a thorough literature review in which we examined the chief (known) correlates with pedestrian motor vehicle crashes.

**Figure 3 ijerph-17-02066-f003:**
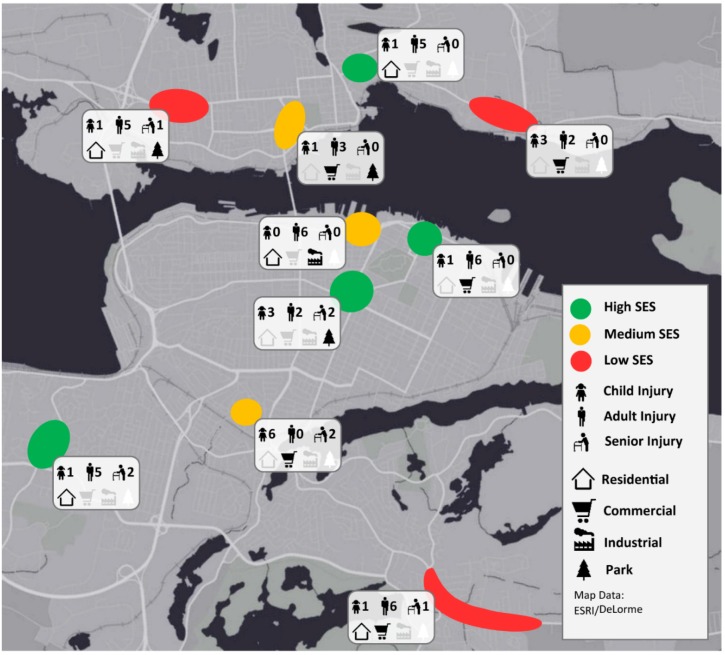
Map of the top ten pedestrian injury hotspot locations showing a range of specific as well as contextual risk correlates. Clear patterns are not observed based on socioeconomic status (SES), age, and land use alone. This indicates the need for further field examination of the individual hotspots in order to understand patterns associated with the built urban environment. Note that this map shows only the top 10 hotspots representing 64 injuries.

**Figure 4 ijerph-17-02066-f004:**
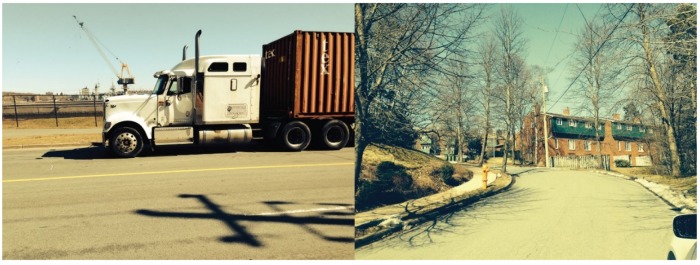
Examples of images that were taken throughout the qualitative scans. The image on the left depicts an industrial area frequented by large trucks, while the image on the right illustrates the low-income house found at one hotspot in an area with blind bends and hills. These factors were associated with increased pedestrian injury risk at three of the hotspots that were surveyed.

**Figure 5 ijerph-17-02066-f005:**
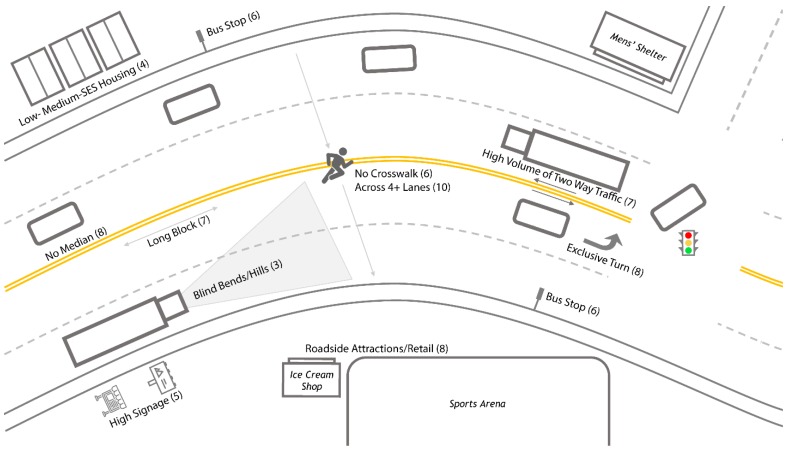
Frequently observed configurations of geographical, socio-economic, and environmental factors present in pedestrian injury hotspots in urban Halifax. The number of hotspots where each factor was observed is listed within parentheses.

**Table 1 ijerph-17-02066-t001:** Injury counts and risk factors by age, sex, and socio-economic deprivation quintile. Note that these are not standardized for population, age, or sex because pedestrian traffic crosses boundaries of neighborhoods and is not exclusive to residence proximity.

Risk Factor	Injury Count	Injury Percent
Age		
0–10	13	7.39%
11–20	41	23.30%
21–30	41	23.30%
31–40	11	6.25%
41–50	15	8.52%
51–60	21	11.93%
61–70	14	7.95%
71–80	10	5.68%
81–90	10	5.68%
Sex		
Male	95	53.98%
Female	81	46.02%
VANDIX SES Ranking Relative to the HRM (Quintiles)		
Q1 (Least Deprived)	41	23.30%
Q2	27	15.34%
Q3	33	18.75%
Q4	28	15.91%
Q5 (Most Deprived)	47	26.70%

**Table 2 ijerph-17-02066-t002:** Frequencies of each risk factor out of ten hotspots.

Risk Factor	Hotspot Prevalence
Long Blocks	7/10
Bus Stops	6/10
Curb Parking	3/10
Cross Walk	6/10
Obstruction	3/10
Exclusive Turn	8/10
Turn Ban	5/10
Bars	2/10
Retail	5/10
School	5/10
Calming	1/10
Median	2/10
Number of Lanes	Mean: 8.8
Signage	High: 5/10
	Medium: 3/10
	Low: 2/10

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
