# Peer review of "Qualitative Field Observation of Pedestrian Injury Hotspots: A Mixed-Methods Approach for Developing Built- and Socioeconomic-Environmental Risk Signatures"

_ijerph, 2020, doi:10.3390/ijerph17062066_

Round 1
Reviewer 1 Report
This paper qualitatively explores the associations between pedestrian injury and built-environment variables through a mixed-methods approach. The Kernel density estimation (KDE) was applied to identify the chief hotspots, the field work and environment scans reveal the unique risk signatures for ten hotspots. This novel idea for mixed-methods approach may be expected to contribute to deeply understand the risks of injury. The authors are dealing with a very interesting work, but it is only poorly theoretical. The works about statistical analysis are few discussed in this paper. Alternatively, the inductive approach is widely used, such as counting the number of bars in each hotspot. Additionally, I have the following comments and suggestions to improve further:
1: Abstract
Please provide full name for abbreviations such as SES, GIS and PVMC.
Is it necessary to use the term “spatial-statistics” in consideration of KDE is not an approach of spatial statistics in GIS. I wonder if “spatial analysis” can be used as an alternative to “spatial-statistics”.
2: Page 2 line 69–70:
“… inherently non-parametric nature of injury data” In this sentence, this expression is few used in previous works; alternatively, it might be more appropriate to say something like “non-parametric nature of models”.
3: Page 2 line 71:
“…identifying risk factors.” this sentence does not provide enough support for why the combination of quantitative and qualitative methodologies is better than individual measures. An appropriate transition that introduces the strengths of qualitative analysis is needed to link these two sentences.
4: Page 4 line 177:
The number of severe pedestrian injury should be added in the revised paper.
5: Page 5 figure 2:
It seems necessary to add a detailed explanation for why this figure is needed.
6: Page 6 line 216-227:
The statistical analysis of SES (i.e. VANDIX matrix) for each DA should be discussed in more detail. Every hotspot may contain several DAs, and hotspot boundaries may cross several DAs, therefore, please provide a clear calculation of SES for each hotspot.
7: Page 7 table 1 and figure 3:
Figure 3 records 76 pedestrian injured, however, in table 1, the number of injury count seems to be greater than this value.
8: Page 9 line 277-278:
“Rather it incorporated … modalities of reporting” in this sentence, “all measures” is vague, and it is necessary to be careful about the expression because with a complex setting, conducting all measures is not possible, such as the measure of psychological changes of injured.
9:Page 10 line 312:
“presence of shelters”, in the results section, it seems to be no discussion about shelter.
10: Page 10 line 339:
An explanation of environmental scans matrix is needed.
11: Page 11 line 362:
“… the effect on injury risk varies geographically” please provide an evidence to support this claim.
12: In line 79 page 2, authors argue that the mixed-methods can bring priori assumptions compared to traditional qualitative research. However, this feature is few discussed in result section and discussion section. Please provide more empirical analysis of this feature for mixed-methods approach.
Reviewer 2 Report
The paper proposes an interesting approach to understand factors contributing to pedestrian injury crashes at high crash areas/locations (called hotspots). The approach relies on different types of information:a checklist to collect road environment features (called environmental scan survey), on-site inspections performed by local experts and socio-economic status in the Dissemination Area (DA) of the hotspot. Compared to approaches mostly based on Police crash data, this approach is more focused at identifying infrastructure and socio-economic factors (hotspot or DA specific) rather then road user behaviour factors (e.g. speeding, distraction, etc). This is the most innovative aspect of the paper in my opinion.
As also stressed by the authors, the main limitation of the approach tested is the underlying assumption that the crashes occurred at a hotspot are caused by the potential factors collected through surveys performed even after 14 years from the crash date. The link between a risk factor and a crash is inferred through the assessments performed by the team.
An important source of information that would fill this gap is data recorded on-scene by the Police investigating the crash.
Specific comments
- Abstract. Please specify the acronims: SES and PMVC.
- 192-193: Specify the meaning of hig/medium/low signage
- Figure 1: This is the methodology adopted, Why in chapter 2.3? I suggest to move it at the beginning of chapter 2.
Round 2
Reviewer 1 Report
In this revision, they responded all my concerning comments, and I think the quality of the paper improved. However, the reference in the introduction is still not enough. Before I address my acceptance of this study, I suggest authors could add following papers focusing on injury analysis and prediction.
[1] Crash injury severity analysis using a two-layer Stacking framework. Accident Analysis and Prevention, Vol.122, 226-238, 2019
[2] Statistical and machine-learning methods for clearance time prediction of road incidents: A methodology review”. Analytic Methods in Accident Research, 2020, Doi: 10.1016/j.amar.2020.100123
[3] “Driving risk assessment using driving behavior data under continuous tunnel environment”., Traffic Injury Prevention. Vol.20, No.8, 807-812, 2019
Author Response
Thanks for these very helpful references. I have added them to page 2, line 69.
All the best, Nadine